# Tail Tales: What We Have Learned About Regeneration from Xenopus Laevis Tadpoles

**DOI:** 10.3390/ijms252111597

**Published:** 2024-10-29

**Authors:** Jessica Lara, Camilla Mastela, Magda Abd, Lenore Pitstick, Rosa Ventrella

**Affiliations:** 1Biomedical Sciences Program, College of Graduate Studies, Midwestern University, Downers Grove, IL 60515, USA; jessica.lara@midwestern.edu (J.L.); camilla.mastela@midwestern.edu (C.M.); magda.abd@midwestern.edu (M.A.); 2Department of Biochemistry and Molecular Genetics, College of Graduate Studies, Midwestern University, Downers Grove, IL 60515, USA; lpitst@midwestern.edu; 3Precision Medicine Program, College of Graduate Studies, Midwestern University, Downers Grove, IL 60515, USA

**Keywords:** regeneration, *Xenopus*, reactive oxygen species (ROS), cell signaling, mechanotransduction, regenerative medicine

## Abstract

This review explores the regenerative capacity of *Xenopus laevis*, focusing on tail regeneration, as a model to uncover cellular, molecular, and developmental mechanisms underlying tissue repair. *X. laevis* tadpoles provide unique insights into regenerative biology due to their regeneration-competent and -incompetent stages and ability to regrow complex structures in the tail, including the spinal cord, muscle, and skin, after amputation. The review delves into the roles of key signaling pathways, such as those involving reactive oxygen species (ROS) and signaling molecules like BMPs and FGFs, in orchestrating cellular responses during regeneration. It also examines how mechanotransduction, epigenetic regulation, and metabolic shifts influence tissue restoration. Comparisons of regenerative capacity with other species shed light on the evolutionary loss of regenerative abilities and underscore *X. laevis* as an invaluable model for understanding the constraints of tissue repair in higher organisms. This comprehensive review synthesizes recent findings, suggesting future directions for exploring regeneration mechanisms, with potential implications for advancing regenerative medicine.

## 1. Introduction

Regenerative medicine has the potential to revolutionize the treatment of various injuries and diseases, like spinal cord injuries, severe burns, damaged heart tissue from heart attacks, and neurodegenerative diseases. These treatments focus on repairing, replacing, or regenerating damaged tissues and organs, which can improve the quality of life for many individuals. Regeneration is a biological process that allows an organism to replace or restore lost or damaged tissues, organs, or limbs [1]. This contrasts with scar formation which is a fast and effective way to repair tissue damage without fully restoring functionality. There are several types of regeneration, including epimorphosis, morphallaxis, compensatory, and stem cell-mediated, each of which is characterized by different mechanisms and levels of complexity [2,3,4,5,6]. Examples of these mechanisms can be found in Table 1. Each of these processes provides unique insights into the potential for regenerative medicine and novel therapies in humans and other organisms lacking regenerative potential. 

Not only does the type of regeneration vary between species, but also the regenerative capability (Figure 1). As the phylogenetic tree progresses to higher-order species like mammals, their regeneration capacity tends to decrease [18,19,20]. Zebrafish can regenerate several organs and appendages such as the heart, pancreas, tail, fin, central nervous system (CNS), and retina [21]. Interestingly, lizards can regenerate their entire tail as adults following a defense mechanism known as caudal autonomy, or voluntary self-amputation of the tail [22]. After tail loss, a wound epithelium is formed followed by the formation of the blastema and then a non-identical replacement tail that lacks vertebrae. In contrast, lizard limb amputation results in wound healing and blastema formation but they are unable to regenerate the missing limb structure [23,24]. Animals like axolotls and salamanders can regenerate a variety of tissues and complex structures into adulthood, whereas frogs undergo regenerative and non-regenerative stages [18,25]. These are all in contrast to mammals that have a more limited regenerative capacity where they can regenerate organs like the liver and pancreas, but not the heart, CNS, or limbs. Mammals also have organs like skin and hair that show partial regenerative capabilities [26,27,28]. As of now, it is still unclear why regeneration has been lost during evolution and why some species over others can regenerate.

Amphibians have mild regenerative capacities and are complex animals that contain a variety of multiple specialized organ systems including the CNS, digestive, and circulatory systems. Because of this, they represent useful model systems to study the dynamic process of regeneration. Specifically, the regenerative capacity of *Xenopus laevis (X. laevis)* can vary throughout development (Figure 2). Tadpoles in the tailbud stage (ST26-ST32), can regrow their tails after tailbud amputation, but it results in shortened, fully developed tails. Originally it was thought that this occurred through regeneration; however, this healing process occurs from displaced trunk tissues, rather than regeneration. When the entire tail-forming region, which includes regions dorsal and anterior to the tailbud, is amputated at this stage, it results in tadpoles that lack tails. At this stage, these multipotent dorsal cells form the ectodermal and mesodermal lineages that are essential for regeneration [30,31,32,33]. However, by stage 40, the tail has completed its development and begins a regeneration-competent phase. But, when entering pre-metamorphosis and feeding begins, between stages 45 and 47 (ST45-ST47), they begin a regeneration-incompetent refractory period where tail amputation does not result in regeneration. They then regain tail regeneration ability several stages later at stage 48 (ST48) [18,30,34,35,36]. 

As tadpoles begin to grow limbs, these structures can regrow following amputation, but this ability is lost, beginning at stage 52 of development; limb regeneration progresses from complete regeneration in pre-metamorphosis, to regeneration with missing digits in metamorphosis, to only spike formation in froglets and adult frogs [30,34,37]. This allows for the unique examination of both regeneration-competent and incompetent stages, as well as the ability to modulate regeneration during these different phases [38]. 

Interestingly, *Xenopus tropicalis* (*X. tropicalis*) does have a refractory period when feeding begins, but it is much more modest than in *X. laevis*, indicating evolutionary divergence between these two highly related *Xenopus* species [39,40]. These differences offer the potential to compare the genetic and molecular underpinnings that control regeneration during this evolutionary divergence. This review will focus on how *X. laevis* tail regeneration has been an invaluable in vivo model system to identify the genetic, molecular, cellular, and developmental mechanisms that control regeneration.

**Figure 2 ijms-25-11597-f002:**
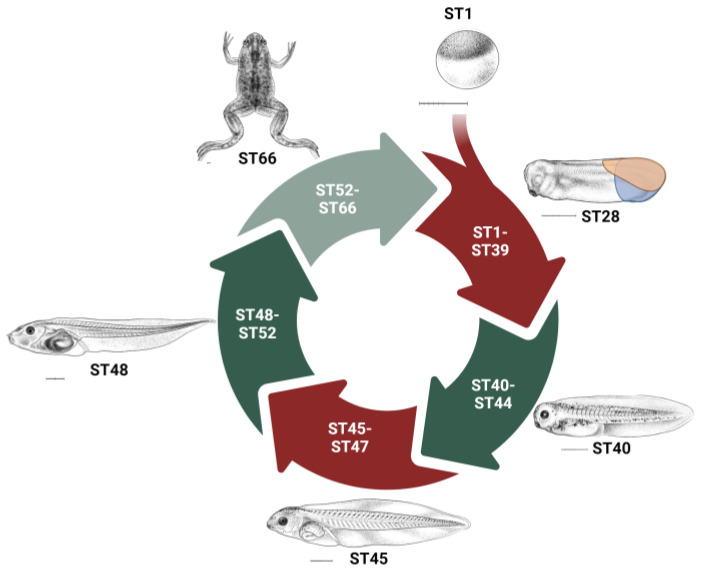
Phases of tadpole regeneration. *Xenopus* have regeneration-competent (green) and regeneration-incompetent (red) phases. Their tails are unable to regenerate in the tail bud stage, but gain regenerative capacity from stage 40 to stage 44. In the tailbud stage, shown at stage 28, if the tailbud is amputated (blue) a shortened tail develops, but if the tail-forming region (orange) is amputated there is no tail regrowth. *Xenopus* then enter a refractory stage in pre-metamorphosis from stage 45 to stage 47, followed by a regenerative competent phase at stage 48. *Xenopus* limbs have maximum regenerative ability at stage 52 and then it declines throughout development. ST: Stage. Scale bar: 1 mm. *Xenopus* illustrations © Natalya Zahn (2022) and sourced from Xenbase (www.xenbase.org RRID:SCR_003280; Accessed on 22 October 2024) [41].

*X. laevis* tadpole tails consist of a spinal cord, somites, notochord, muscle, and skin; these structures can completely regenerate within seven to fourteen days after amputation. Their tail regeneration undergoes epimorphic regeneration, which can be broadly divided into three distinct phases including the early stage which is characterized by scar-free wounding healing and inflammation (0–6 h post-amputation; hpa), the intermediate stage which is characterized by proliferation and the formation of the blastema (6–48 hpa), and then the late phase of differentiation and regenerative outgrowth (2–7 days post-amputation; dpa) [18]. Amputation induces an immediate response that is characterized by inflammation and the generation of a protective wound epithelium that covers the damaged site. This is followed by the formation of a blastema under the wound epithelium that contains a mass of undifferentiated cells that are crucial for regeneration. These cells then proliferate to provide cells for the regenerating tissues. Gradients of signaling molecules provide positional information within the blastema to organize the cells and ensure that the regenerating tail forms in the correct orientation and structure. These cells then differentiate into specific tissue types needed for the tail, such as muscle, notochord, spinal cord, and skin, and integrate with the existing tail tissues. Lastly, the regenerating tail elongates as new tissues are formed and integrated until the regenerating tail achieves full functionality, including the restoration of movement, blood flow, and neural connections [8,42]. Each of these steps is coordinated by a range of signaling mechanisms to ensure that proper structure and function are fully re-established (Table 2).

## 2. Signaling in Regeneration

### 2.1. Proinflammatory Signaling from Regeneration Organizing Cells (ROCs) 

The transforming growth factor-β (TGFβ) superfamily of cytokines includes TGFβs, bone morphogenic proteins (BMPs), activins, and inhibins, among many others, have been shown to play important roles in migration, invasion, and remodeling of the extracellular matrix [43]. Because of this, the role of these secreted factors in regeneration has been deeply investigated. These ligands elicit signaling in a similar manner where they bind to a specific set of type I and type II kinase receptors resulting in the phosphorylation, activation, and translocation into the nucleus of SMAD transcription factors. SMADs can bind to SMAD-binding elements (SBEs) in DNA and with additional transcription factors, they can modulate transcription. This TGFβ family of cytokines can also operate in a SMAD-independent pathway where the activated kinase receptors directly activate other downstream pathways like Rho/ROCK, RAS/Raf/MEK/ERK, PI3K, and NF-κB [43].

Specifically, TGFβ signaling is essential for tail regeneration. Following amputation, TGFβ1 transcripts are increased at the tail edge, and knockout or inhibition of TGFβ1 impairs proliferation and subsequent tail regeneration [44,45]. This is in part through the TGFβ-induced activation of SMAD, which results in JunB transcription. JunB can cooperate with Inhibinβα to induce regenerative outgrowth through fibroblast growth factor receptor (FGFR) signaling [46,47,48]. Single-cell RNA sequencing has identified a population of cells as regeneration-organizing cells (ROCs) that migrate to the wound end following tail amputation in *X. laevis* tails. These ROCs express a variety of ligands known to be required for regeneration including TGFβ, as well as FGFs, BMPs, Wnts, and Notch, which can activate their corresponding receptors in the nearby progenitor cells (PCs) to increase proliferation [36,49,50,51]. Interestingly, the grafting of posterior tail-bud tissues containing ROCs to the trunk of *X. laevis* induces ectopic outgrowths, reminiscent of tail-like structures, further highlighting how these ligand-secreting ROCs are an essential signaling center for regeneration (Figure 3) [36]. 

### 2.2. Oxidative Eustress in Tail Regeneration

Reactive oxygen species (ROS), the most notable being hydrogen peroxide (H_2_O_2_), are derived from molecular oxygen and formed through redox reactions or electron excitation. Many external stimuli, including physical stressors, cytokines, and growth factors, can lead to the production of ROS, which then can alter downstream protein activity and signaling. Intracellular concentrations of ROS are important signaling molecules and can lead to various cellular processes including proliferation, differentiation, migration, angiogenesis, and stress adaptation when in a state of oxidative eustress. However, there is a fine balance in these levels as too much can lead to oxidative distress resulting in growth arrest and cell death [52]. 

Previous studies during *X. laevis* tail regeneration have shown that ROS plays an essential role. After tail amputation, there is an influx of extracellular oxygen and ROS remain high during the first four days of tail regeneration [53,54]. Another potential source of ROS is due to the melanocortin receptor 4 (Mc4r) as this receptor is required to activate the production of ROS and sustain ROS levels during the regeneration of *X. laevis* limbs [55]. Inhibition of ROS levels during amputation-induced regeneration with NADPH oxidase inhibitors, a ROS antioxidant, and knockdown of the NOX subunit *cyba* impair proliferation and tail regeneration. This outcome is likely due to two essential downstream ROS pathways. First, decreased ROS is accompanied by reduced Wnt/β-catenin signaling at the wound site, suggesting that lower levels of ROS impair the migration of ROCs to the amputation plane [53]. Additionally, oxygen influx and ROS generation create a hypoxic environment in the regenerative bud and sufficient regeneration is dependent on the stabilization of the master mediator of hypoxia, HIF-1α. Although ROS do not directly stabilize HIF-1α, they do so indirectly by maintaining a hypoxic environment (Figure 4). Interestingly, stabilization of HIF-1α during the *X. laevis* regeneration-incompetent stage was sufficient to induce regeneration [54]. Although it is known that HIF-1α can modulate ROCs indirectly through ROS, at this time, it is unknown if HIF-1α plays a direct role in the migration of ROCs to the amputation plane.

Another potential signaling pathway that is dependent on ROS production for regeneration is NF-κB. Normally, NF-κB proteins are sequestered in the cytoplasm by IκB, but upon activation of various membrane receptors, like TLRs, TNFRs, and IL-1R, IκB becomes degraded, allowing NF-κB to translocate to the nucleus and act as a cofactor for the transcription of target genes [56]. These target genes include a variety of cytokines, chemokines, and cell cycle regulators that play roles in inflammation, cell proliferation, and survival [57]. Following *X. laevis* tail amputation, NF-κB translocates to the nucleus in a ROS-dependent manner to induce the transcription of target genes, including *cox-2*, *nox2*, and *nox4* (Figure 4)*. nox2* and *nox4* code for protein subunits of the NADPH oxidase complex, so the increased expression of these genes could provide a feed-forward loop to supply sustained ROS levels required for regeneration [58]. 

There is also a crosstalk between NF-κB and other signaling pathways, including Wnt and Notch, both of which act downstream of BMP [59]. Activation of Wnt is required for regeneration through β-catenin-induced FGF expression, as well as through the activation of JNK [50,60]. Additionally, Wnt/β-catenin can promote NF-κB-induced gene expression [61]. Notch signaling is also required for proper tail regeneration and the Notch intracellular domain (NICD) can promote NF-κB activity by preventing its cytoplasmic sequestration [35,62]. Collectively, it is likely that FGF expression, NICD-induced gene expression, and NF-κB-induced proinflammatory signaling work cooperatively to promote the migration of ROCs to the amputation site and induce the differentiation of PCs to promote regeneration.

In addition to oxygen influx, amputation stimulates the release of calcium. This is due to internal calcium stores as inhibition of ryanodine receptors in the endoplasmic reticulum (ER) impairs regeneration [63,64]. One potential mechanism for ER-induced calcium release is through the transmembrane protein, c-answer. This receptor was identified in evolution studies that aimed to identify genes that had been lost in warm-blooded animals that lack regenerative capacities, compared to cold-blooded regenerating animals. c-answer interacts with the purinergic G protein-coupled receptor P2Y1 and activation of this receptor by ADP initiates the Gα_q_/PLC signaling cascade ultimately leading to calcium release by the ER [65,66]. These calcium transits occur in immature muscle cells hours after amputation and begin to decrease after 24 h and are required for muscle satellite cell activation and muscle cell precursor proliferation in the regenerating tail [63,64].

### 2.3. Mechanotransduction Signaling in Tail Regeneration

Mechanotransduction is the ability of cells to sense and respond to mechanical stimuli to elicit a biochemical response that results in intracellular changes, oftentimes through cell signaling [67]. One signaling pathway that has been known to be central in mechanotransduction is the Hippo/YAP pathway. This pathway has often been studied in the development of cancer, largely due to it being involved with cell development, proliferation, and apoptosis. It is regulated by the transcriptional coactivator Yes-associated protein (YAP) which functions in both the nucleus and the cytoplasm depending on its phosphorylation status. A variety of upstream regulators, including tight junctions, G-protein coupled receptors (GPCRs), integrins, receptor tyrosine kinases (RTKs), and adherens junctions, play a role in controlling YAP activation. When YAP is phosphorylated, it will remain in the cytoplasm, while unphosphorylated YAP will localize to the nucleus and bind to TEAD transcription factors to regulate the expression of genes involved in proliferation, EMT and cell migration, and cell plasticity, depending on the involvement of other transcription factors. The upstream regulators can control YAP phosphorylation directly, or by modulating the activation of the hippo kinases, MST1/MST2, which, when active, promote YAP phosphorylation [68]. YAP localization also responds to mechanical stimuli, where low tension results in YAP phosphorylation, cytoplasmic sequestration, and possible proteasomal degradation, while high tension prevents YAP phosphorylation allowing for nuclear translocation [69]. Due to the vast amount of tissue remodeling and tension changes that occur during regeneration, one would expect that YAP is essential to regulate regeneration. 

During *X. laevis* tadpole tail regeneration, YAP is expressed in the regenerating tailbud, and active YAP transcription is required for tail regeneration in a cell-autonomous manner. Loss of YAP through a dominant-negative YAP or dominant-negative TEAD impairs tail regeneration. This is likely due to a lack of proliferation and increased ectopic apoptosis [70]. Similar results were seen in the requirement of YAP activity in limb regeneration in *X. laevis*, axolotl limb, and mitten crabs [71,72,73]. These results not only imply that YAP-induced transcription is required for proper regeneration but also highlight the importance of mechanotransduction during this process. 

Extensive tissue remodeling results in a dynamic mechanical environment, which will ultimately affect downstream signaling. The temporal dynamics of this process have not been well characterized. Also, the interaction of YAP/TEAD with different transcription factors can alter transcriptional outputs, so a better understanding of the gene regulatory network of YAP/TEAD will provide better insight into this process. Moreover, ROS have been shown to alter the microenvironment through extracellular matrix remodeling to make an environment that is more conducive to migration and invasion. However, ROS-induced extracellular matrix degradation would likely result in a less stiff environment therefore preventing nuclear YAP accumulation [74]. Also, evidence in heart regeneration has shown that ROS results in YAP cytoplasmic retention and degradation [75]. So, although YAP-induced transcription is required for tail regeneration, it is unlikely that its activation is directly through increased extracellular matrix (ECM) stiffness or induction of ROS, as these mechanisms would most likely lead to decreased YAP activation. However, there is a fine-tuned balance between ROS levels and beneficial or detrimental cellular outcomes [76,77,78]. Because of this, cytostatic ROS levels may be required for increasing ECM stiffness, YAP activation, and regeneration, whereas higher ROS levels and oxidative stress may have the opposite effect. Accordingly, there are possible prospective upstream YAP regulators that are essential for controlling this signaling pathway and counteracting mechanisms that may inhibit it. 

### 2.4. Metabolic Alterations to Meet the Demands of Regeneration

Since the refractory stage in *Xenopus* occurs as the animals shift from feeding on maternal yolk stores to independent feeding, it has been suggested that nutrient mobilization and metabolism play a significant role in regeneration. The importance of nutrients has been seen in *X. tropicalis*. In this model system that does not have as drastic a refractory period as *X. laevis*, the inhibition of nutrient metabolism by inhibiting mTOR signaling reduces regeneration and regeneration competence could be restored during the refractory period by feeding the tadpoles [40]. It has also been proposed that there is a metabolic shift during regeneration that increases glucose internalization and shifts its metabolism towards anabolic pathways. After amputation, there is an increase in the expression of glucose transporters as well as *g6pd* and *hif-1α* [79]. Glucose-6-phosphate dehydrogenase (G6PD) catabolizes the rate-limiting step of the pentose phosphate pathway (PPP), which can generate a variety of biomolecules important for DNA and protein synthesis and increased levels of HIF-1α can suppress mitochondrial metabolism [80]. Additionally, ROS can inhibit the glycolic enzyme pyruvate kinase M2 (PKM2), diverting additional glucose to the PPP. Collectively, these metabolic changes may help to generate the additional biomolecules required to regenerate the tissue [81]. However, understanding the precise mechanisms and outcomes of these metabolic changes remains relatively unexplored [79]. 

## 3. Epigenetic Control of Tail Regeneration

To ensure that proper signaling pathways are employed during regeneration, epigenetic control during regeneration is a finely tuned process. Epigenetics can alter gene expression through chromatin organization, histone and DNA modifications, and transcriptional regulation [82]. Histone acetylation and methylation of different lysine residues can alter chromatin structure; histone acetylation by histone acetyltransferases (HATs) creates permissive chromatin-promoting gene expression, whereas histone deacetylation by proteins such as histone deacetylases (HDACs) generates repressive chromatin that reduces gene expression [83]. During *X. laevis* tail regeneration, histone H3 lysine 9 acetylation (H3K9Ac) peaks 24 h after amputation, most notably in the notochord. Additionally, this epigenetic change is dependent on the release of ROS following amputation, suggesting that ROS signaling facilitates the expression of genes required for regeneration, like *shh*, *wnt*, and *fgf*, through H3K9Ac in the notochord [84]. However, HDAC activity is also required for tail regeneration, highlighting the complexity of histone dynamics [85,86]. 

These opposing outcomes could suggest that a dynamic histone state is required for regeneration, where genes can be turned on and off in a time- and space-dependent manner. This idea can further be supported by gene expression changes in neural progenitor cells (NPCs) during tail regeneration in *X. tropicalis*. After amputation, NPCs prioritize migration and tubule morphogenesis, followed by neural differentiation, and lastly by proliferation [87]. Genes regulating these cellular processes need to be temporally controlled, and the inability to properly regulate their dynamics could impair tissue regeneration. Additionally, there can be crosstalk between different histone modifications [88]. So, altering global histone acetylation could affect other histone modifications, including histone phosphorylation, ubiquitination, and methylation, resulting in the modified expression of many genes, ultimately leading to a loss of regeneration.

## 4. Innate Immune Responses in Tail Regeneration

### 4.1. The Contribution of the Microbiome to Regeneration 

Proper microbial colonization of the wound site helps to initiate the innate immune response. When *X. laevis* tadpoles are raised in an environment that lacks Gram-negative bacteria, there is a loss of tail regeneration. However, a lack of antibiotics or reconstitution of the media with E-coli or lipopolysaccharides (LPS) can rescue this response [58,89]. This response is, in part, due to LPS-induced activation of the toll-like receptor 4 (TLR4) on resident macrophages, but other pathogen-associated molecular patterns (PAMPs) and TLRs are likely involved in this immune response [89]. Identification of these other PAMPs and TLRs will provide insight into how the innate immune system can be modified to alter regeneration, as well as predict how the microbiome can be modulated to increase regenerative capacity.

The skin microbiomes of *X. laevis* vary greatly between different sibships and environments but are mostly dominated by Gram-negative proteobacteria, although they are highly variable at the genus level [89,90]. As expected, treatment with gentamycin greatly reduced the number of bacteria as well as the proportion of Gram-negative bacteria, but it also increased the microbial diversity on the skin. This corresponded to decreased tail regeneration, further highlighting the correlation between Gram-negative bacteria and regenerative capacity [89]. Since the regeneration-incompetent refractory period in *X. laevis* development is only about a day long, it is unlikely that the resident microbiome undergoes large changes during this time. However, the identification of changes in bacterial secretion during this time may reveal potential substances that prevent regeneration. Additionally, correlating the makeup of the resident microbiome with regeneration capacity may identify novel signaling pathways, like that of LPS, that can regulate regeneration. 

### 4.2. Amputation-Induced Recruitment of Immune Cells During Regeneration

Due to the importance of NF-κB, SMAD, and HIF-1α, cytokines likely play an important role in initiating these downstream signaling pathways. The interleukin-6 family member interleukin-11 (IL-11) and the interleukin-11 receptor subunit alpha (IL-11RA) are required for *X. laevis* tail regeneration. IL-11 is abundantly expressed in the blastema and maintains the tissue progenitor cells during this process [91,92,93]. However, in the absence of a blastema like in mammals, injury-induced upregulation of IL-11 causes inflammation and tissue fibrosis leading to epithelial dysfunction and organ failure [94,95]. Therefore, IL-11 may be essential in eliciting a pro-inflammatory response to induce regeneration; however, the lack of a blastema to respond to this cytokine results in the perpetuation of a pro-inflammatory and fibrotic response.

Following injury, innate immune cells are recruited to the injury site and this response is essential for regeneration [96]. Specifically, the loss of macrophage function impairs regeneration in fish, amphibians, and mammals [97,98,99,100]. Cell tracking in *X. laevis* has shown that neutrophil-like cells migrate to the injured site within 20 minutes, which is then followed by the recruitment of macrophage-like cells after an hour [96]. This is at a similar time of ROS activation, suggesting that these two processes may be linked [53,101].

Additionally, myeloid cells are required for tissue remodeling and ROC mobilization to the amputation plane. During regeneration, there are two populations of myeloid cells, including the inflammatory myeloid cells and the reparative myeloid cells. During regeneration, there is an increase in the reparative myeloid cell population, whereas the regeneration-incompetent stage is characterized by sustained inflammatory myeloid activity indicating that regeneration incompetence may be due to a failure in suppressing the pro-inflammatory state [102]. Similarly, the immune responses differ between the regeneration-competent and -incompetent stages. During regenerative phases, there is a transient induction of inflammation; however, in the refractory stage this inflammation is chronic [103]. The inhibitory effect of inflammation on regeneration can be counteracted, as treatment of *X. laevis* with immunosuppressants during the regeneration-incompetent phase can restore regenerative abilities resulting in ECM remodeling and mobilization of ROCs to the amputation plane [102,103]. 

## 5. Remodeling of the ECM During Regeneration

Hyaluronan (HA) is a glycosaminoglycan ECM protein that is synthesized by hyaluronan synthase and can be highly crosslinked with other ECM components like proteoglycans and interacting cell surface receptors like CD44 and the receptor for hyaluronic acid-mediated motility (RHAMM) [104]. Following amputation, there is increased expression of the HA pathway, including hyaluronan synthase 2 (HAS2), hyaluronidase 2 (Hyal-2), CD44, and RHAMM, in the regenerative bud, and depletion of HA results in a loss of mesenchymal cell proliferation and regeneration [105]. Additionally, the inhibition of HA decreases ROC mobilization to the amputation plane [102]. Loss of HA causes decreased expression of Wnt signaling and inhibition of the Wnt-antagonist, GSK3β, can restore regeneration in the absence of HA synthesis [105]. Because of this, HA synthesis following amputation is likely an upstream component that coordinates the activation of Wnt-target genes. These downstream factors can then promote proliferation in the regenerative bud and be secreted to direct the migration of ROCs to the amputation plane.

The importance of ECM remodeling is further seen from the identification of a population of cells known as the regeneration-initiating cells (RICs) that form in the regenerative bud. The regenerative bud becomes highly enriched in this cell population that emerges from the basal epidermal population a few hours after amputation and they are characterized by genes involved in ECM degradation and remodeling including *mmp1*, *mmp8*, and *mmp9*. Removal of the regenerative bud that contains these RICs or knockdown of these cells results in a lack of ROC migration to the wound edge followed by fibrosis and scarring, rather than regeneration [106]. Following amputation, there is also an increase in the reparative myeloid cells that express MMPs [102]. Together, this myeloid cell population and the RICs likely play an essential role in remodeling the ECM in the regenerative bud to promote the migration of cells required for regeneration to occur.

## 6. Conclusions

The study of *X. laevis* tadpole tail regeneration offers crucial insights into the cellular, molecular, and developmental processes underlying tissue repair. The work described in this review highlights the critical roles of ROS, mechanotransduction, epigenetic regulation, and various essential signaling pathways in orchestrating regeneration. Yet, despite these advances, several important gaps remain that must be addressed in future research to understand the mechanisms of regeneration lost throughout evolution and how these mechanisms can be harnessed for regenerative medicine. 

While the regenerative capabilities of *Xenopus* have been well-characterized in relation to specific signaling molecules such as TGFβ, BMPs, and FGFs, the intricacies of how these pathways interact in different temporal and spatial contexts remain unclear. The regeneration process can be broken down into three main phases including wound healing, followed by blastema formation, and then finally regenerative outgrowth. A summary of the regulatory pathways discussed in the review and their role in the different stages of regeneration are outlined in Table 2. Regeneration is a tightly coordinated process that involves multiple layers of signaling, from immediate inflammatory and wound-healing responses to long-term tissue differentiation. A more detailed understanding of the crosstalk between these pathways, particularly in the context of ROS production, would further refine our understanding of how these mechanisms operate synergistically to enable regeneration. Additionally, much of the work on regeneration has focused on the wound healing and blastema stages of regeneration and a gap still remains in understanding the signaling mechanisms that are required for tissue patterning and tail outgrowth. 

The identification of ROCs as essential mediators of tail regeneration was a landmark in understanding the dynamic process of regeneration of the tail epithelium [36]. The signaling landscape that governs these cells’ behavior, including how they respond to external mechanical and biochemical cues, is still not fully understood. Investigating how these cells migrate, proliferate, and differentiate into specific tissue types could provide a clearer picture of their role in regeneration. Additionally, there could be subpopulations of ROCs, that play discrete roles in coordinating the time and space coordination of regeneration. Similarly, there could be ROC populations in other tissues like the spinal cord and muscle that are required for the regeneration of that tissue. If this is the case, communication between these ROC populations is likely required to ensure proper regeneration of a fully functional tail. Lastly, understanding the evolutionary biology of these ROCs could give insight into potential therapeutic mechanisms that may be used to restore regeneration in regeneration-incompetent species. 

The review also highlights the importance of mechanotransduction, epigenetic regulation, and metabolism in regeneration, yet significant questions remain. While YAP signaling has been implicated in the regeneration process, the exact mechanics of how tissue remodeling and changes in mechanical tension influence gene expression and cellular behavior require further investigation. Additionally, the dynamic role of histone modifications and their regulation during regeneration points to the complexity of epigenetic control, suggesting that temporally and spatially fine-tuned chromatin states are necessary for successful tissue repair. More research into the interaction between different histone marks and how they influence regeneration at different stages would be beneficial. Lastly, the role of metabolism in regeneration remains relatively underexplored. It would be expected that regenerating all the tissues in the tail would require a significant amount of energy and biomolecules. A metabolic shift towards anabolic pathways could be one mechanism utilized to support the demands of tissue repair [79]. Although there have been some proposed mechanisms for this shift, the precise signaling that controls this shift remains unknown. Metabolomic studies will likely be useful in better understanding how metabolism is altered in *X. laevis* tail regeneration, which could be useful in identifying the metabolic requirements for regeneration and evaluating if some of these requirements have been lost during evolution. 

The comparative aspect of regeneration between species poses one of the most intriguing questions in evolutionary biology: Why do some animals regenerate while others do not? Understanding the genetic and molecular differences between regenerative and non-regenerative species could shed light on the evolutionary trade-offs that may have led to the loss of regenerative capabilities in higher vertebrates. Bioinformatic genetic screens have been performed to identify genes that have been lost as animals have evolved from cold-blooded regenerating animals to warm-blooded non-regenerating animals [65]. Similar bioinformatics studies comparing the differences in the epigenome, proteome, and metabolome between cold-blooded and warm-blooded animals may offer insight into the evolutionary loss of regeneration. 

Significant progress has been made in understanding the molecular underpinnings of *Xenopus* tail regeneration, but the complexity of this process demands further investigation. Future research will likely focus on uncovering the nuances of signaling pathways, the role of metabolism, and the evolutionary factors that govern regenerative capacity. Identifying molecular differences in mammals compared to species like *Xenopus* could provide valuable clues for unlocking latent regenerative potential in humans. With this goal in mind, an adult *X. laevis* regeneration-incompetent hindlimb model was developed and has proved to be a promising model system for investigating regenerative medicine therapies. In this model, a pro-regenerative multidrug treatment delivered by a bioreactor to amputated *X. laevis* hindlimbs was successful in promoting regrowth, tissue repatterning, and regeneration [107]. With the identification and increased understanding of the timing of novel pathways, this adult *X. laevis* hindlimb model system can be used to explore mechanisms for promoting regeneration in normally regenerative-incompetent appendages. As our knowledge of these mechanisms deepens, the potential for translating these insights into regenerative medicine applications, such as treatments for spinal cord injuries or heart disease, becomes increasingly promising.

## Figures and Tables

**Figure 1 ijms-25-11597-f001:**
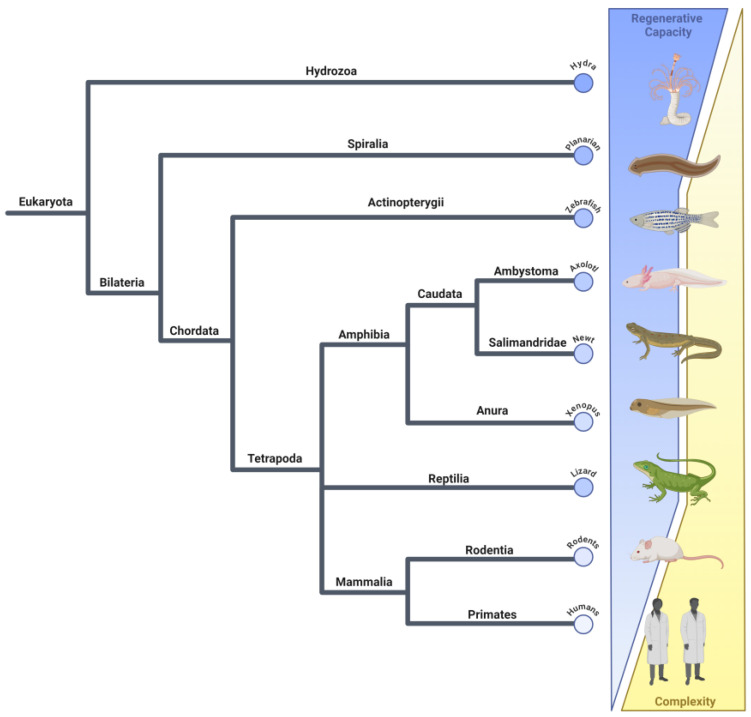
There is an inverse relationship between regenerative capacity and animal complexity. The regenerative capacity of animals has decreased throughout the evolution of more complex organisms. The lines in the phylogenetic tree are not scaled with regard to time. Animals are organized by iTOL v6 and ordered by regenerative capacity [29]. Image created with BioRender.com, accessed on 22 October 2024.

**Figure 3 ijms-25-11597-f003:**
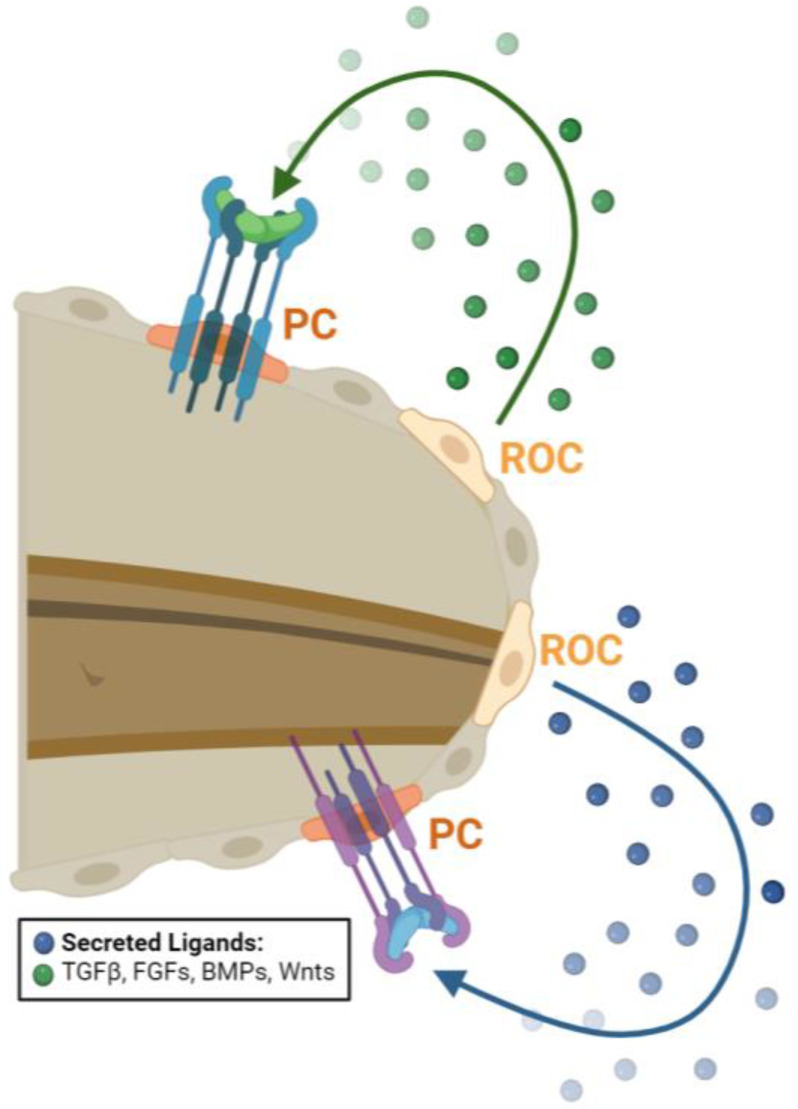
Regeneration-organizing cells (ROCs) are required for tail regeneration. Following amputation, ROCs, shown in yellow, migrate to the amputation plane and secrete ligands like TGFβ, FGFs, BMPs, and Wnts that promote proliferation in progenitor cells (PCs), shown in orange. Image created with BioRender.com.

**Figure 4 ijms-25-11597-f004:**
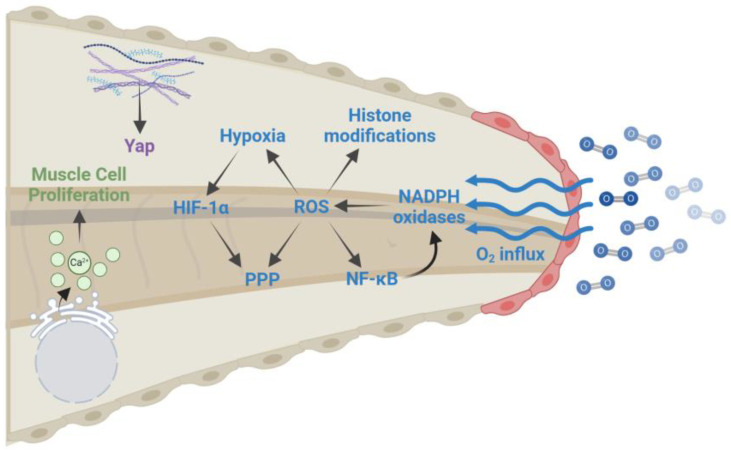
Signaling pathways involved in *X. laevis* tail regeneration. Amputation causes oxygen influx at the damaged site (red cells) which activates NADPH oxidases to generate ROS that (1) activate NF-κB which can create a feed-forward loop by increasing the expression of NADPH oxidase subunits, (2) create a hypoxic environment resulting in the stabilization of HIF-1α, (3) alter histone modifications, and (4) divert glucose to the PPP along with HIF-1α. Additionally, the transcriptional cofactor, YAP, responds to mechanical stimuli and is required for tail regeneration. Lastly, calcium release from the endoplasmic reticulum is required for muscle cell proliferation during regeneration. Image created with BioRender.com.

**Table 1 ijms-25-11597-t001:** Different types of regeneration with specific examples.

Type of Regeneration	Definition	Examples
**Epimorphosis**	Involves the formation of a mass of undifferentiated cells, known as a blastema, at the site of injury. These cells differentiate into the various cell types needed to regrow the lost structure [2,7].	Amphibians can regenerate entire limbs, including bones, muscles, nerves, and skin [2,7]. *Xenopus* tadpoles can regenerate their tails through the formation of a blastema that differentiates into the various tissues of the tail [8].
**Morphallaxis**	Involves the reorganization of existing tissues without significant cell proliferation. This process typically results in the direct transformation of existing cells into a new structure [2,3].	*Hydra* can regenerate its entire body from a small fragment by reorganizing its existing cells to form a complete organism [2,9]. Planarians can regenerate from small body fragments through a combination of morphallaxis and epimorphosis, involving both reorganization and proliferation of cells [2,3,10].
**Compensatory**	Occurs when differentiated cells divide but maintain their original function. There is no formation of a blastema, and the regeneration typically restores function rather than form [4].	The mammalian liver can regenerate lost tissue through compensatory hypertrophy and hyperplasia. Hepatocytes grow and divide to restore the liver’s mass and function without forming a blastema [11,12].
**Stem cell-mediated**	This involves the activation and differentiation of stem cells for regeneration, either through undifferentiated stem cells or tissue-specific progenitors, without the formation of a blastema [6].	The mammalian intestinal epithelium is continuously regenerated by stem cells located in the crypts of the intestinal lining [13,14]. In the hematopoietic system, blood cells are regenerated from hematopoietic stem cells in the bone marrow, which continuously produce new blood cells throughout an organism’s life [15]. Human skeletal muscle can regenerate through the activation of satellite cells, which are muscle-specific stem cells that proliferate and differentiate to repair muscle fibers [16]. Newts can regenerate the lens of their eyes when the cells from the iris dedifferentiate and proliferate to form a new lens [17].

**Table 2 ijms-25-11597-t002:** Major regulatory signaling networks throughout the regenerative process.

Regeneration Stage (Time After Amputation)	Dominant Regulatory Pathways
**Wound healing** **(0–6 hpa)**	ROS production: Oxygen influx at the damage site leads to increased ROS levels that are essential for regeneration. ROS levels remain elevated for several days after amputation and are required for activating NF-κB, creating a hypoxic environment, altering histone modifications, and diverting glucose to the pentose phosphate pathway.
	Calcium signaling: Calcium is released from the ER following injury and induces the activation of muscle satellite cells and the proliferation of muscle progenitor cells.
	Inflammatory response: The recruitment of innate immune cells, including macrophages, neutrophils, and myeloid cells, to the amputation site is required for regeneration.
	ECM remodeling: ECM remodeling genes are expressed in the RICs that emerge hours after amputation. RICs combined with upregulation of the HA pathway promote the migration of ROCs to the wound edge.
	TGFβ signaling: This signaling pathway is required for the proper formation of the wound epidermis following amputation.
**Blastema formation** **(6–48 hpa)**	Growth factor signaling: Additional growth factors, like FGFs, BMPs, and Wnts, are secreted by ROCs to promote the proliferation of progenitor cells in the blastema.
	Anti-inflammatory response: The inflammatory response must be dampened for regeneration to occur. Chronic inflammation leads to a loss of regenerative capabilities.
	Epigenetic changes: H3K9Ac peaks one day after amputation and likely facilitates the expression of genes required for cell proliferation and regenerative outgrowth. However, there are a variety of epigenetic changes in which the dynamics are not well known and likely regulate other epigenetic modifications.
**Regenerative outgrowth** **(2–7 dpa)**	Hippo/YAP signaling: Yap is expressed several days after amputation and maintains the survival of neural progenitors during the growth of the regenerating tail.

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
