# Peer review of "Tail Tales: What We Have Learned About Regeneration from Xenopus Laevis Tadpoles"

_ijms, 2024, doi:10.3390/ijms252111597_

Round 1
Reviewer 1 Report
Comments and Suggestions for Authors
A nice review of Xenopus tail regeneration concentrating on the early stages and not the later cell morphogenesis stages. It begins with a consideration of wider evolutionary aspects of regeneration across invertebrates and vertebrates. This consideration generates most of my comments below.
Fig. 1 and lines 44 – 56. 1) why have reptiles been left out of this consideration? They are famous for regenerating their tails, they can regenerate CNS tissue, produce a blastema on the limb (but not regenerate any further). Thus they are better at regenerating than adult Anurans. I think they deserve to be there.
2) line 46 onwards “zebrafish have high regenerative capacity……… whereas amphibians have a more limited repertoire of regenerating organs”. Not true – axolotls and salamanders are as good as zebrafish if not better.
Line 54 ‘mammals also have organs like skin and hair that show partial regenerative capabilities [18].’ Why is the citation for mammalian skin and hair regeneration a review of Xenopus regeneration (Phipps et al., 2020). Very odd.
Line 62 what does ‘mild complexity’ mean?
Reviewer 2 Report
Comments and Suggestions for Authors
The present review discusses Xenopus laevis tail regeneration after amputation as a model for regenerative medicine. The review is concise, very well written, and very interesting from a regenerative medicine perspective. While this paper should be accepted for publication, this reviewer believes that the addition of some discussion and clarification points would enrich the manuscript. These suggestions are encouraged given the short extension of the review, which leaves room for some topics interesting to the regenerative medicine community:
1) In sections 2, 3 and 4, the relevance in each stage of development of the tadpole is not discussed. Are these molecules and regulatory pathways equally important in all stages? Are there differences between those stages with different regenerative capabilities? A table to summarize differences between different stages would be very helpful.
2) A description of the advancement in wound healing events in time at higher detail (including shift of molecules and pathways), comparing at least regenerative and non-regenerative stages of the tadpoles, would also be beneficial
3) Is there any literature on X.laevis production of commonly known inflammatory cytokines like IL-6, IL-8, IL-12, IL1b, or TNFa during regeneration? These cytokines are very intimately related to the pathways described (Nfkb, Ikb, SMAD, HIFa, etc.). Conversely, is there any literature / studies on pro-remodeling cytokines, factors (IL-10, IL-4, etc).
4) The review should also discuss the production of ECM components and how these relate with the described pathways. Concretely, wound healing in mammals proliferative phase is characterized by the deposition of matrix components (collagen type I and III and fibronectin mainly) but in an unorganized manner, where the late shift to the remodeling phase makes this matrix to become scar tissue rather than functional matrix. Are these events equivalent in the model discuss? In both regenerative and non-regenerative stages?
5) Similarly, a little deeper discussion should be made about the innate immune system. Is the characteristic shift to pro-remodeling phenotypes in innate cells happening? Which cells?
Minor comments:
6) In figure 2 Yellow and purple colors are very faint and difficult to distinguish
7) In some parts of the review YAP is typed as Yap. Please revise.
8) Authors should add some references or examples of the model employed for studies in specific applications of regenerative medicine, if available.
